# Asymmetric formal C–C bond insertion into aldehydes via copper-catalyzed diyne cyclization

Cui-Ting Li[1], Lin-Jun Qi[1], Li-Gao Liu[1], Chang Ge[1], Xin Lu ![ORCID][1] ✉, Long-Wu Ye ![ORCID][1,2] & Bo Zhou ![ORCID][1] ✉

The formal C–C bond insertion into aldehydes is an attractive methodology for the assembly of homologated carbonyl compounds. However, the homologation of aldehydes has been limited to diazo approach and the enantioselective reaction was rarely developed. Herein, we report an asymmetric formal C–C bond insertion into aldehydes through diyne cyclization strategy. In the presence of Cu(I)/SaBOX catalyst, this method leads to the efficient construction of versatile axially chiral naphthylpyrroles in moderate to excellent yields with good to excellent enantioselectivities. This protocol represents a rare example of asymmetric formal C–C bond insertion into aldehydes using non-diazo approach. The combined experimental and computational mechanistic studies reveal the reaction mechanism, origin of regioselectivity and stereoselectivity. Notably, the chiral phosphine ligand derived from synthesized axially chiral skeleton was proven to be applicable to asymmetric catalysis.

Preparation and derivatization of aldehydes represent the fundamental transformations in organic synthesis[1]. While the traditional nucleophilic addition onto aldehydes has been well-studied over the past century, modern strategies that enable the formal insertion into aldehydes have attracted increasing interest recently[2–6]. Compared with the formal C–H bond insertion of diazo compounds into aldehydes, namely, the Roskamp reaction (path a, Fig. 1a)[7–13], the related formal C–C bond insertion has been much less explored[14–27]. The reaction starts with the nucleophilic addition of the diazoalkyl carbon onto aldehyde, followed by 1,2-migration of R[1] group and the extrusion of N$_2$, thus providing one-carbon homologated aldehyde (path b, Fig. 1a). For instance, Hossain[14–17], Kanemasa[18], Kirchner[19,20], Maruoka[21], Ryu[22], and others[23–27] independently investigated the C–C bond insertion of diazoacetates into aldehydes for the generation of homologated aldehydes or corresponding enols. In 2013, the only catalytic asymmetric example of formal C–C bond insertion of diazoesters into aldehydes was reported by Ryu group, through the incorporation of a chiral oxazaborolidinium ion catalyst[22].

Despite the efficacy of these established approaches, the problematic product distribution between C–C and C–H insertion products, as well as the hazardous and explosive nature of diazo compounds should not be ignored, which could significantly limit the application of this strategy. To address these limitations, new catalytic systems operated by non-diazo methodology for the homologation of aldehydes become imperative.

Alternatively, we suppose that the highly reactive carbonyl ylides[28–35] could be potentially applicable to the formal C–C bond insertion into aldehydes. In recent years, special attention were drawn to the formation of carbonyl ylides via the reaction of aldehydes with metal carbenes, which provided efficient access to a variety of 1,3-dioxolanes[36–47] and epoxides[48–50]. Despite these achievements, the formal aldehyde homologation was not observed in the reaction of aldehydes with metal carbenes. Mechanistically, it originates from the intrinsic requirement of electron-withdrawing substituents on these metal carbenes, which could inhibit the desired carbon migration. Hence, it is vital to figure out a carbene-like intermediate which is not

[1]State Key Laboratory of Physical Chemistry of Solid Surfaces, Key Laboratory of Chemical Biology of Fujian Province, and College of Chemistry and Chemical Engineering, Xiamen University, Xiamen 361005, China. [2]State Key Laboratory of Organometallic Chemistry, Shanghai Institute of Organic Chemistry, Chinese Academy of Sciences, Shanghai 200032, China. ✉e-mail: xinlu@xmu.edu.cn; zhoubo@xmu.edu.cn

constrained to electron-deficient substituents, such as a donor/donor carbene-like intermediate (Fig. 1b).

To circumvent the inherent challenges in applying carbonyl ylides into the formal C–C bond insertion, we envisaged that vinyl cation could be a prospective option[51–53], due to the unique carbene-like reactivity. Our laboratory has been engaged in the facile diyne cyclization for the generation of vinyl cations, which were considered as useful donor/donor carbene-like intermediates. The diyne cyclization enabled varieties of asymmetric transformations via remote control of enantioselectivity[54–59], including intramolecular aromatic C(sp²)–H functionalization[54], vinylic C(sp²)–H functionalization[55], cyclopropanation[54], [1,2]-Stevens-type rearrangement[56],

intermolecular annulations with styrenes[57] and ketones[58], as well as oxidation and X–H insertion[59]. Inspired by these results and our recent study on developing ynamide chemistry for heterocycle synthesis[60–65], we envisioned that the vinyl cation intermediate generated from diyne cyclization should be a suitable precursor of carbonyl ylide with electron-rich substituents, thus facilitating the formal C–C bond insertion into aldehydes. Moreover, the asymmetric version could be realized through remote-stereocontrol of chiral copper catalyst. Herein, we disclose the development of such an asymmetric formal C–C bond insertion into aldehydes through copper-catalyzed diyne cyclization, which offers a versatile entry to homologated pyrrylaldehydes and axially chiral naphthylpyrroles[66] (Fig. 1c). To the best of our

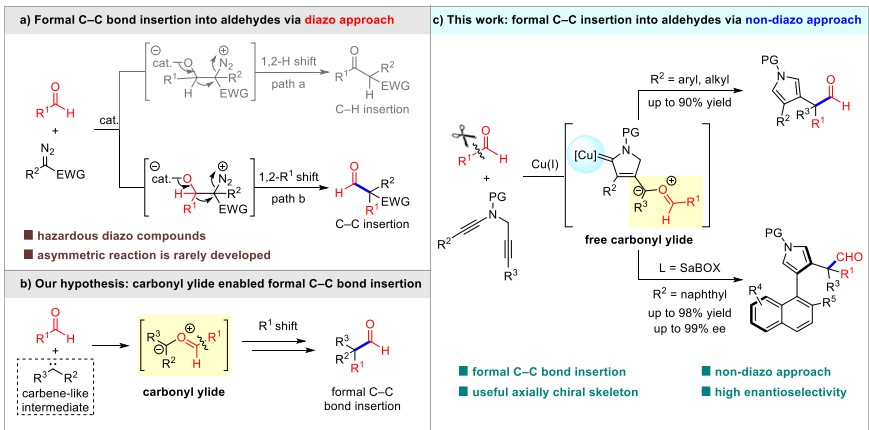

**Fig. 1 | Catalytic formal C–C bond insertion into aldehydes. a** Formal C–C bond insertion into aldehydes via diazo approach. **b** Our hypothesis: carbonyl ylide enabled formal C–C bond insertion. **c** This work: formal C–C bond insertion into aldehydes via non-diazo approach.

## Table 1 | Optimization of reaction conditions[a]

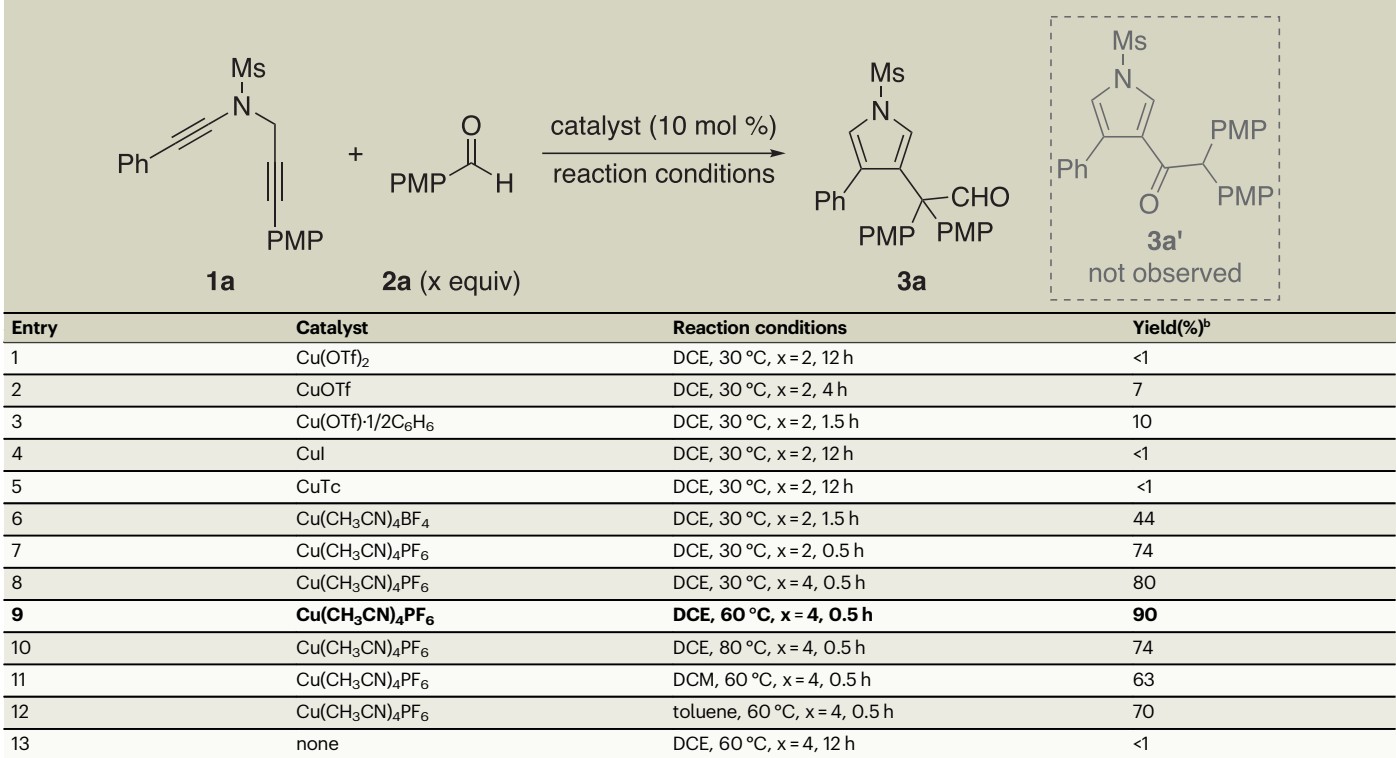

| Entry | Catalyst | Reaction conditions | Yield(%)[b] |
|---|---|---|---|
| 1 | Cu(OTf)₂ | DCE, 30 °C, x = 2, 12 h | <1 |
| 2 | CuOTf | DCE, 30 °C, x = 2, 4 h | 7 |
| 3 | Cu(OTf)·1/2C₆H₆ | DCE, 30 °C, x = 2, 1.5 h | 10 |
| 4 | CuI | DCE, 30 °C, x = 2, 12 h | <1 |
| 5 | CuTc | DCE, 30 °C, x = 2, 12 h | <1 |
| 6 | Cu(CH₃CN)₄BF₄ | DCE, 30 °C, x = 2, 1.5 h | 44 |
| 7 | Cu(CH₃CN)₄PF₆ | DCE, 30 °C, x = 2, 0.5 h | 74 |
| 8 | Cu(CH₃CN)₄PF₆ | DCE, 30 °C, x = 4, 0.5 h | 80 |
| **9** | **Cu(CH₃CN)₄PF₆** | **DCE, 60 °C, x = 4, 0.5 h** | **90** |
| 10 | Cu(CH₃CN)₄PF₆ | DCE, 80 °C, x = 4, 0.5 h | 74 |
| 11 | Cu(CH₃CN)₄PF₆ | DCM, 60 °C, x = 4, 0.5 h | 63 |
| 12 | Cu(CH₃CN)₄PF₆ | toluene, 60 °C, x = 4, 0.5 h | 70 |
| 13 | none | DCE, 60 °C, x = 4, 12 h | <1 |

[a]Reaction conditions: **1a** (0.05 mmol), **2a** (0.1–0.2 mmol), catalyst (0.005 mmol), solvent (1 mL), 30 °C to 80 °C, 0.5–12 h, in vials.
[b]Determined by ¹H NMR spectroscopy using 1,3,5-trimethoxybenzene as the internal standard. Ms = methanesulfonyl, PMP = 4-methoxyphenyl, DCE = 1,2-dichloroethane.

knowledge, this protocol represents a rare example of asymmetric formal C−C bond insertion into aldehydes using non-diazo approach.

# Results

## Screening of reaction conditions

To examine the feasibility of the proposed formal C−C bond insertion strategy, N-propargyl ynamide **1a** was selected as the precursor of vinyl cation to react with 4-methoxybenzaldehyde **2a** in the presence of various copper catalysts (Table 1). At the outset, the comparison of Cu(OTf)$_2$ and CuOTf revealed that Cu(I) could be the alternative catalyst for this transformation (entries 1−2). Further evaluation of other Cu(I) catalysts, including Cu(OTf)·1/2C$_6$H$_6$, CuI, and CuTc, resulted in the one-carbon homologated pyrrylaldehyde **3a** in low yields (entries 3−5). Gratifyingly, the switch of catalyst to Cu(I) complexes with CH$_3$CN as the ligand, such as Cu(CH$_3$CN)$_4$BF$_4$ and Cu(CH$_3$CN)$_4$PF$_6$ brought significant improvement of yields (44−74%) and reaction rates (entries 6−7). To our delight, the reaction could be further promoted by increasing the equivalent of aldehyde **2a** and using higher temperature, which allowed the formation of homologated aldehyde **3a** in 80−90% yields (entries 8−9). In addition, the reaction was proved to be less efficient when performed at 80 °C (entry 10) or in other solvents such as DCM and toluene (entries 11−12). The reaction failed to give any conversion in the absence of a copper catalyst (entry 13). It's notable that the competitive formal C−H bond insertion into aldehyde (ketone byproduct **3a'**) was not observed in the above conditions, showing the excellent regioselectivity for the C−C bond cleavage (for more conditions see Supplementary Table 1).

## Reaction scope study

With the optimal reaction conditions in hand (Table 1, entry 9), the substrate scope of the formal C−C bond insertion was then investigated. In general, N-propargyl ynamides **1** containing functional groups with different electronic properties reacted well with arylaldehydes **2** to give the corresponding one-carbon homologated aldehydes **3** (Fig. 2). Diynes **1** with different sulfonyl protecting groups, including Ms (**3a**), Ts (**3b**), MBS (**3c**), SO$_2$Ph (**3d**) and Bs (**3e**), reacted smoothly to deliver pyrrylaldehydes in 70−90% yields. A wide array of aryl-substituted N-propargyl ynamides (R$^1$ = Ar) bearing both electron-withdrawing and -donating groups on the benzene ring at *para* (**3f**−**3j**) and *meta* (**3k, 3l**) positions, were all compatible with the formal C−C insertion. More challenging alkyl-substituted N-propargyl ynamide containing cyclopropyl substituent was also proved to be a competent substrate (**3m**). Interestingly, it was found that a range of aryl-substituted N-propargyl ynamides (R$^2$ = Ar) were suitable substrates to generate desired pyrrylaldehydes **3n**−**3r** in 33−79% yields. The substrate scope of arylaldehydes **2** was next investigated under the optimized reaction conditions. Phenyl aldehydes equipped with a series of functional groups, including Et (**3s**), OTIPS (**3t**), NMe$_2$ (**3u**) and OMe (**3v**), successfully produced the corresponding homologated aldehydes in moderate to good yields. It should be noted that, the transformation also exhibited good tolerance for heterocycles, such as furan (**3w**) and indole (**3x**). The structure of product **3b** was confirmed by X-ray diffraction. Attempts to extend the reaction to non-substituted benzaldehyde and alkyl aldehydes resulted in low yields (for more details see Supplementary Fig. 1), probably due to the lower nucleophilicity of these aldehydes.

**Fig. 2 | Substrate scope for the formal C−C bond insertion.** Reaction conditions: **1** (0.2 mmol), **2** (0.8 mmol), Cu(CH$_3$CN)$_4$PF$_6$ (0.02 mmol), DCE (4 mL), 60 °C, 0.5−1.5 h, in vials; yields were those for the isolated products. [a]**2b** (2 mmol), 50 °C. PG = protecting group, MBS = 4-methoxybenzenesulfonyl, Bs = 4-bromobenzenesulfonyl.

**Table 2 | Optimization of reaction conditions for the atroposelective formal C–C bond insertion[a]**

| Entry | L | Reaction conditions | Yield (%)[b] | ee (%)[c] |
|---|---|---|---|---|
| 1 | L1 | DCE, 30 °C, x = 2, 1 h | 76 | 40 |
| 2 | L2 | DCE, 30 °C, x = 2, 1 h | 75 | 10 |
| 3 | L3 | DCE, 30 °C, x = 2, 1 h | 80 | 63 |
| 4 | L4 | DCE, 30 °C, x = 2, 1 h | 82 | 88 |
| 5 | L5 | DCE, 30 °C, x = 2, 1 h | 84 | 73 |
| 6 | L6 | DCE, 30 °C, x = 2, 1 h | 82 | 85 |
| 7 | L7 | DCE, 30 °C, x = 2, 1 h | 77 | 86 |
| 8 | L4 | DCE, 30 °C, x = 4, 1 h | 80 | 87 |
| 9 | L4 | DCM, 30 °C, x = 2, 1 h | 83 | 83 |
| 10 | L4 | PhCF₃, 30 °C, x = 2, 1 h | 83 | 63 |
| **11** | **L4** | **DCE, 0 °C, x = 2, 18 h** | **88** | **92** |
| 12 | L4 | DCE, –10 °C, x = 2, 48 h | 80 | 92 |

[a]Reaction conditions: **4a** (0.05 mmol), **2h** (0.1–0.2 mmol), Cu(CH₃CN)₄PF₆ (0.005 mmol), **L** (0.006 mmol), NaBArF₄ (0.006 mmol), solvent (1 mL), –10 °C to 30 °C, N₂, 1–48 h, in Schlenk tubes.
[b]Measured by ¹H NMR using 1,3,5-trimethoxybenzene as the internal standard.
[c]Determined by HPLC analysis. NaBArF₄ = sodium tetrakis[3,5-bis(trifluoromethyl)phenyl]borate.

After establishing a general and reliable method for this formal C–C bond insertion into aldehydes, we proposed to develop an asymmetric version of this copper-catalyzed aldehyde homologation, for the enantioselective construction of quaternary carbon stereocenters. However, after multiple attempts using an array of chiral ligands, we found the stereocontrol on the quaternary carbon remains unsatisfied (<10% ee). Inspired by our recent work involving axial chirality[59,67–69], we considered that diynes preinstalled with sterically demanding groups might react with arylaldehydes via atroposelective formal C–C bond insertion to produce axially chiral arylpyrroles (Table 2). Firstly, naphthyl diyne **4a** and arylaldehyde **2h** were chosen as the model substrates to conduct this atroposelective transformation. The proof-of-concept experiment was performed with catalytic amount of Cu(CH₃CN)₄PF₆/(R)-MeO-BIPHEP (**L1**) and NaBArF₄ in DCE at 30 °C, which delivered the axially chiral naphthylpyrrole **5a** in good yield albeit with moderate ee value (entry 1). However, the

enantioselectivity couldn't be further improved after screening of other bisphosphine ligands (for more conditions see Supplementary Table 2). Gratifyingly, a survey into Tang's sidearm-modified bisoxazoline ligands (SaBOX)[70] led to significant improvement in the enantioselectivity. As listed in Table 2, the modification of sidearms and arenes on SaBOX ligands (**L2**–**L7**) afforded **5a** in 75–84% yields with 10–88% ees (entries 2–7). Afterwards, the screening of **2h**'s equivalent and solvents failed to promote the reaction (entries 8–10). On further optimization of reaction conditions by varying the reaction temperature, we determined the following optimal conditions to evaluate the substrate generality: **4a** (1 equiv) was treated with **2h** (2 equiv) in the presence of Cu(CH₃CN)₄PF₆ (10 mol %), **L4** (12 mol %) and NaBArF₄ (12 mol %) in DCE at 0 °C for 18 h, and the desired axially chiral product **5a** was obtained in 88% yield with 92% ee (entry 11).

The substrate scope of this atroposelective formal C–C bond insertion was next explored under the optimized reaction conditions

(Table 2, entry 11). As depicted in Fig. 3, the reaction proceeded smoothly to deliver the corresponding axially chiral naphthylpyrroles **5** in generally moderate to excellent yields with good to excellent enantioselectivities. First, diynes with different sulfonyl protecting groups reacted well to deliver the homologated aldehydes **5a**–**5e** in 70−89% yields with 86−93% ees. The reaction also demonstrated good tolerance towards various substituents (such as alkyl, OMe, TMS) on different positions of naphthalene ring and furnished the desired axially chiral naphthylpyrroles **5f**–**5n** in 40−98% yields with 80−99% ees. A variety of N-propargyl ynamides **4** and aldehydes **2** bearing identical aryl substituents (Ar$^1$ = Ar$^2$), were competent substrates to afford the homologated aldehydes in moderate to good yields with 87−94% ees (**5o**–**5s**). Besides phenyl aldehydes, other heterocyclic aldehydes (Ar$^1$ ≠ Ar$^2$) underwent the regioselective formal C−C bond insertion to provide the corresponding naphthylpyrroles in moderate yields with excellent enantiocontrol (**5t**–**5v**). It should be mentioned that the diastereocontrol involving the quaternary carbon stereocenter is still challenging at this stage. Therefore, this protocol provides an efficient pathway for the construction of divergent potentially useful axially chiral naphthylpyrroles.

## Synthetic applications

To explore the synthetic utility of this atroposelective formal C−C bond insertion, a gram-scale reaction was first carried out, leading to the desired product **5a** in 85% yield with 92% ee (Fig. 4). Furthermore, the axially chiral naphthylpyrrole **5a** can readily undergo various transformations to access diverse structural motifs. For example, the

sulfonyl-protecting group on pyrrole moiety could be easily removed to provide free pyrrole product **6**, which was further diversified to dibrominated pyrrole **7** and Boc-protected pyrrole **8**. Alternatively, the aldehyde moiety facilitated a wide range of derivatizations, including 1,2-reduction, reductive amination, condensation, and Wittig olefination, thus resulting in axially chiral naphthylpyrroles **9**–**12** efficiently. Note that these transformations all proceeded without erosion of enantiopurity. The absolute configuration of **11** was unambiguously confirmed by X-ray diffraction.

Moreover, an axially chiral phosphine ligand based on enantioenriched naphthylpyrrole **12** was successfully prepared to prove the practicability of this methodology (Fig. 5a). Naphthylpyrrole **12** could undergo deprotection and bromination to give monobrominated free pyrrole **13** in good yield. Subsequent methylation and base-promoted phosphination furnished the desired axially chiral phosphine ligand **14**. Notably, phosphine **14** could be used as a suitable ligand to realize the silver-catalyzed enantioselective [3 + 2] cycloaddition of azomethine ylide **15** and maleimide **16**, as well as the palladium-catalyzed enantioselective allylic alkylation of 1,3-diphenylallyl acetate **18** and dimethyl malonate **19** (Fig. 5b). These results demonstrate that the constructed enantioenriched axially chiral skeleton is capable of inducing the chirality in asymmetric synthesis.

## Mechanistic investigations

To elucidate the mechanistic details, several control experiments were conducted (Fig. 6). The reaction of selected diyne **1s** with aldehyde **2e** under optimized reaction conditions at 30 °C allowed the formation of

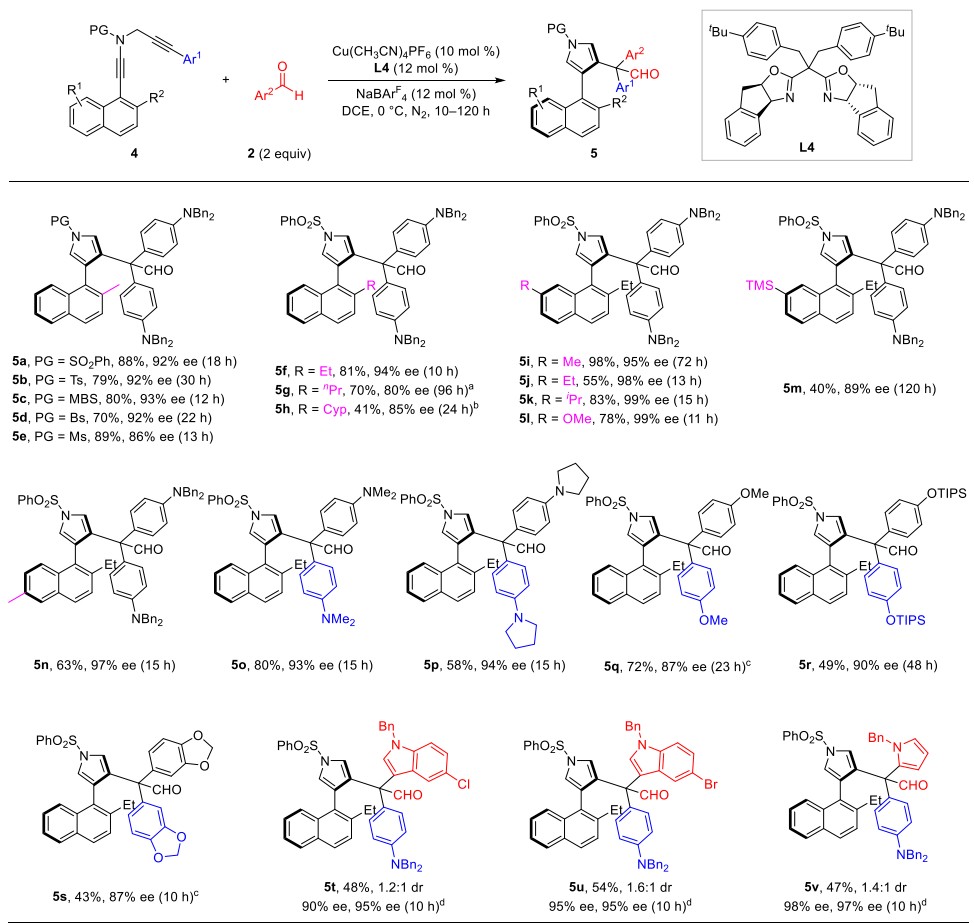

**Fig. 3 | Substrate scope for the atroposelective formal C−C bond insertion.**
Reaction conditions: **4** (0.1 mmol), **2** (0.2 mmol), Cu(CH$_3$CN)$_4$PF$_6$ (0.01 mmol), **L4** (0.012 mmol), NaBAr$^F_4$ (0.012 mmol), DCE (2 mL), 0 °C, N$_2$, 10−120 h, in Schlenk tubes; yields were those for the isolated products; ees were determined by HPLC analysis. $^a$20 °C. $^b$**L3** was used instead of **L4**. $^c$25 °C. $^d$−10 °C. Cyp = cyclopentyl.

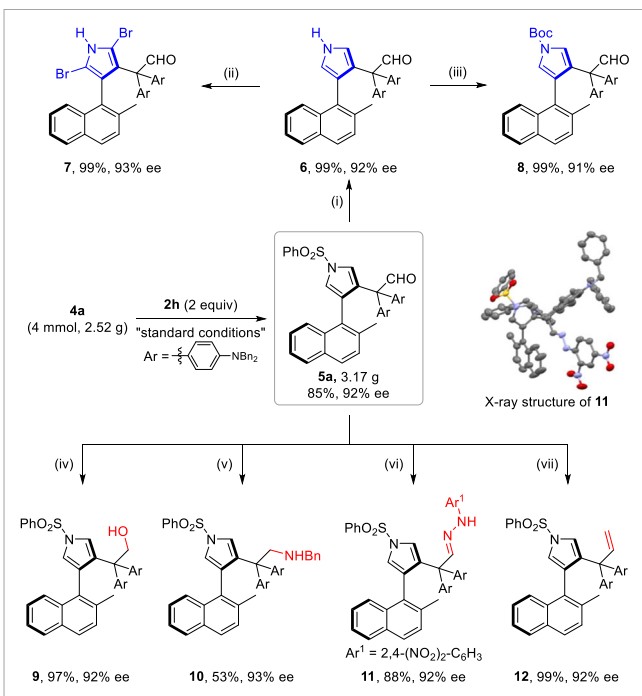

**Fig. 4 | Gram-scale reaction and further transformations.** Reagents and conditions: (i) KOH (3 equiv), EtOH/THF = 1/1, 50 °C, 0.5 h. (ii) NBS (2.1 equiv), THF, 0 °C, 0.5 h. (iii) Boc₂O (4 equiv), 4-DMAP (0.5 equiv), MeCN, 40 °C, 1 h. (iv) NaBH₄ (2 equiv), MeOH/THF = 1/1, rt, 0.5 h. (v) BnNH₂ (4 equiv), NaBH₃CN (4 equiv), AcOH (3.3 equiv), MeOH/THF = 1/1, rt, 2 h. (vi) DNPH (5 equiv), EtOH/DCE/AcOH = 1/1/1, 80 °C, 1.5 h. (vii) Ph₃PMeBr (2 equiv), ⁿBuLi (2 equiv), THF, −78 °C–rt, 2 h.

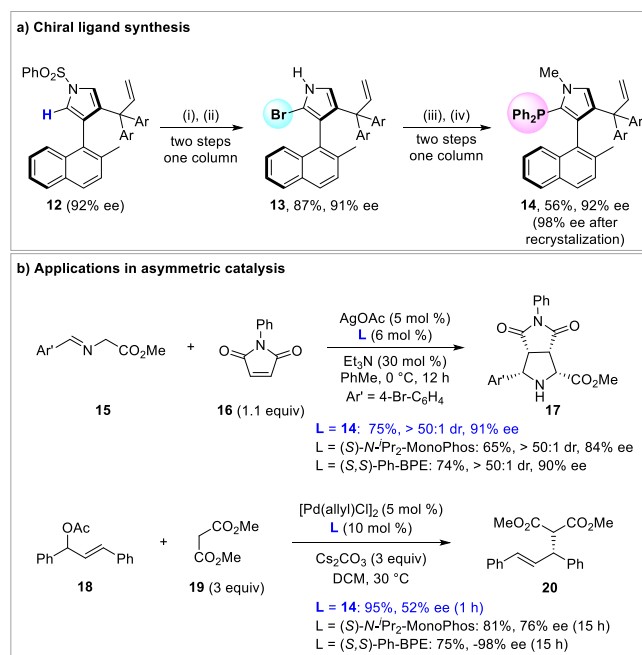

**Fig. 5 | Synthetic applications. a** Chiral ligand synthesis. **b** Applications in asymmetric catalysis. Reagents and conditions: (i) KOH (3 equiv), EtOH/THF = 1/1, 50 °C, 1 h. (ii) DBMDH (0.6 equiv), THF, −78 °C, 10 min. (iii) MeI (4 equiv), NaH (6 equiv), DMF, 40 °C, 1 h. (iv) ClPPh₂ (5 equiv), ⁿBuLi (5 equiv), THF, −78 °C, 40 min.

desired product **3y** as well as 1,3-dioxolane **3y'**[36–47]. Further treatment of **3y'** with Cu(CH₃CN)₄PF₆ at 60 °C delivered the desired product **3y** with the exclusion of aldehyde **2e** in similar yields. These results indicated that the 1,3-dioxolane **3y'** should be the key intermediate and subsequent ring-opening step is presumably involved in the transformation.

On the basis of the above experimental results, density functional theory (DFT) calculations were performed to illustrate the mechanism for the formal C−C bond insertion. As displayed in Fig. 7a, the reaction begins with the coordination of BOX-ligated copper species to diyne **4f** to form intermediate **A**, followed by intramolecular atroposelective cyclization to give vinyl cation intermediate **B** via transition state **TS$_A$** with a free energy barrier of 8.6 kcal/mol. Nucleophilic addition of aldehyde onto vinyl cation intermediate **B** affords vinyl copper intermediate **C** which is considered as the resonance form of carbonyl ylide. In line with the developed reactions of aldehydes with metal carbenes[36–47], the second nucleophilic addition of aldehyde onto intermediate **C** delivers copper carbene intermediate **D** bearing a 1,3-dioxolane moiety. Subsequent Lewis base-assisted 1,4-proton transfer and demetallation[54–59] take place to afford 1,3-dioxolane intermediate **E**, which has been isolated in the above control experiments. Then, a Lewis-acid promoted ring-opening of copper-coordinated 1,3-dioxolane intermediate **F** leads to carbocation intermediate **G** with a free energy barrier of 20.7 kcal/mol, which is the rate-determining step. Finally, the desired naphthylpyrrole (*R*)-**5f** forms via 1,2-aryl migration. In another alternative pathway (blue line), the coordination of copper with another oxygen atom on 1,3-dioxolane unit could result in intermediate **F'**, which undergoes further ring-opening to give the ketone product **5f'**. A comparison of the intermediates **F** and **F'** indicates that the copper-coordinated intermediate **F'** has much higher free energy compared with intermediate **F** (−50.1 kcal/mol versus −59.9 kcal/mol), which might due to the difficult coordination of copper onto sterically

hindered oxygen. Therefore, the generation of ketone product **5f'** is less likely and the regioselective formation of homologated aldehyde (*R*)-**5f** is thermodynamically preferred. In addition, the detailed reaction mechanism for the formation of the enantiomer (*S*)-**5f** was also calculated (for more details see Supplementary Figure 5).

The origin of enantioselectivity was also theoretically explored by using SaBOX coordinated Cu(I) complex in the enantioselectivity-determining intramolecular cyclization step (**A → B**). As shown in Fig. 7b, the free energy of transition state **TS$_A$-(*R*)** (leading to the major enantiomer) is predicted to be 2.2 kcal/mol lower than that of **TS$_A$-(*S*)** (leading to the minor enantiomer) and the theoretically predicted enantioselectivity matches well with the experimental ee value (96.6% versus 94%). Inspection of the structures of transition states shows that **TS$_A$-(*S*)** has a shorter C···C distance than **TS$_A$-(*R*)** for the bond-forming position (2.07 Å versus 2.09 Å), and the ligand **L4** on **TS$_A$-(*S*)** is closer to naphthyl group than **TS$_A$-(*R*)** (closest distance: 2.41 Å versus 2.47 Å). These results suggest that **TS$_A$-(*S*)** has stronger steric repulsion and lower stability. Therefore, the observed enantioselectivity is dominated by steric effects.

## Discussion

In summary, an asymmetric formal C−C bond insertion into aldehydes has been developed through a copper-catalyzed diyne cyclization strategy, which provides a convenient entry for the construction of axially chiral naphthylpyrroles. This methodology has significant advantages over the existing diazo approach, including the avoidance of hazardous and explosive reagents, as well as high regioselectivity. Importantly, the reaction represents a rare example of asymmetric formal C−C bond insertion into aldehydes using a non-diazo approach. The scalability and further derivatizations lead to the divergent synthesis of axially chiral skeletons, which were proven to be potentially applicable as chiral ligands to asymmetric catalysis. Furthermore, experimental and computational mechanistic studies have been carried out to understand the reaction mechanism, origin of regioselectivity and stereoselectivity. Although still in its infancy, we believe the findings shown here will shed lights on the development of aldehyde

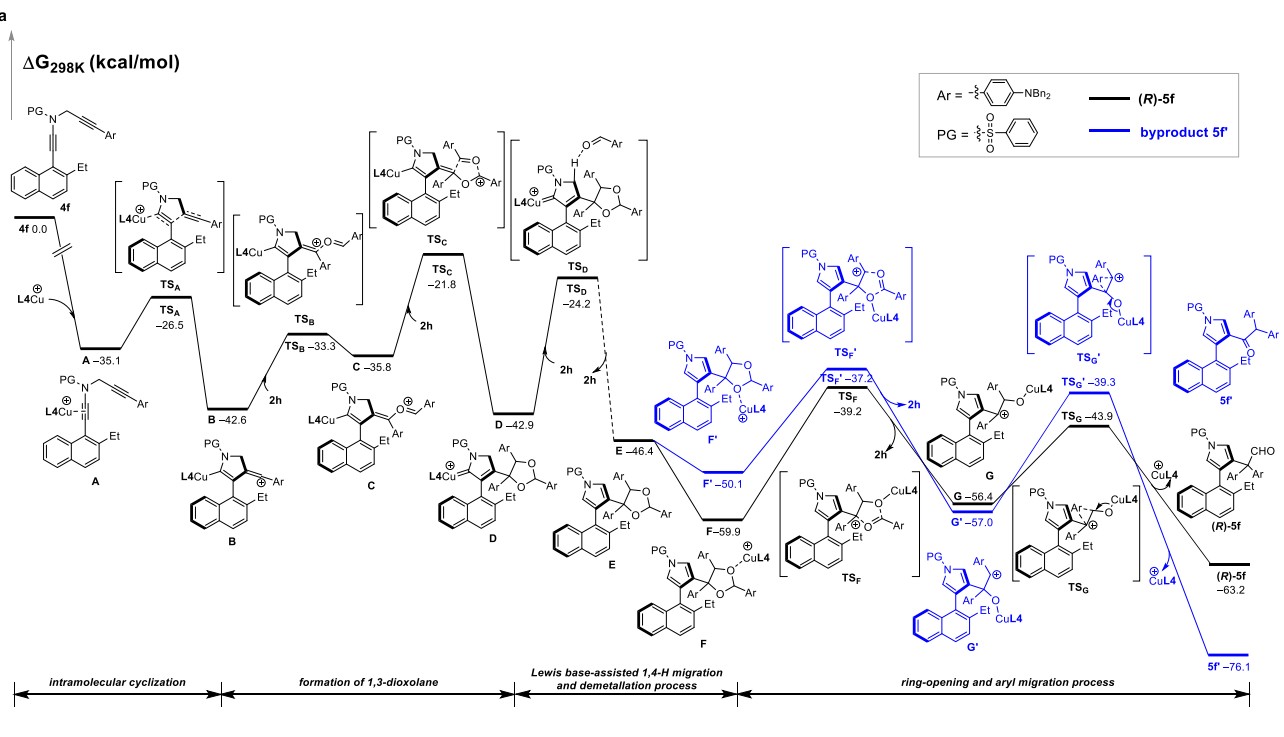

**Fig. 6 | Control experiments. a** Isolation of 1,3-dioxolane intermediate. **b** Reaction of 1,3-dioxolane intermediate in standard conditions.

**Fig. 7 | Plausible reaction mechanism. a** Plausible reaction pathway. **b** The geometries and relative free energies of the transition states **TS_A-(R)** and **TS_A-(S)** with the chiral ligand. Relative free energies (ΔG, in kcal/mol) of all the transition states and intermediates were computed at the SMD(dichloroethane)-B3LYP-D3/def2TZVP//SMD(dichloroethane)-B3LYP-D3/6-31 G(d)/LANL2DZ level of theory.

homologation. Efforts to expand the aldehyde scope and the discovery of more efficient catalytic systems are the topics of the ongoing investigation.

## Methods

### General procedure for the synthesis of pyrrylaldehydes 3

To the solution of *N*-propargyl ynamide **1** (0.2 mmol) and arylaldehyde **2** (0.8 mmol) in DCE (4 mL) was added $Cu(CH_3CN)_4PF_6$ (7.6 mg, 0.02 mmol). The resulting mixture was then stirred at 60 °C for 0.5–1.5 h and the progress of the reaction was monitored by TLC. Upon completion, the resulting mixture was concentrated under reduced pressure and purified by column chromatography on silica gel (eluent: hexanes/ethyl acetate) to afford the desired pyrrylaldehyde **3**.

### General procedure for the synthesis of axially chiral naphthylpyrroles 5

To a dry 10 mL Schlenk tube charged with a stir bar were added $Cu(CH_3CN)_4PF_6$ (3.8 mg, 0.01 mmol), **L4** (7.5 mg, 0.012 mmol), $NaBAr^F_4$ (10.6 mg, 0.012 mmol) and DCE (1 mL) sequentially under $N_2$ atmosphere. The solution was stirred at room temperature for 2 h. After cooling to 0 °C, the solution of *N*-propargyl ynamide **4** (0.1 mmol) and arylaldehyde **2** (0.2 mmol) in DCE (1 mL) was added into the reaction dropwise. The resulting mixture was stirred at 0 °C for 10–120 h and the progress of the reaction was monitored by TLC. Upon completion, the resulting mixture was concentrated under reduced pressure and purified by column chromatography on silica gel (eluent: hexanes/ethyl acetate) to afford the desired axially chiral naphthylpyrrole **5**.

## Data availability

Data for the crystal structures reported in this paper have been deposited at the Cambridge Crystallographic Data Centre (CCDC) under the deposition numbers CCDC 2235694 (**3b**), 2235698 (**11**). Copies of these data can be obtained free of charge via www.ccdc.cam. ac.uk/data_request/cif. For $^1H$, $^{13}C$ and other nuclear magnetic resonance (NMR) spectra of compounds in this manuscript and details of the synthetic procedures as well as more reaction conditions screening, see Supplementary Information. All other data supporting the findings of this study, including experimental procedures and compound characterization, are available within the paper and its Supplementary Information files, or from the corresponding authors on request. The coordinates of the optimized structures in this study are provided in the Source Data file. Source data are provided with this paper.

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

## Acknowledgements

We are grateful for financial support from the National Natural Science Foundation of China (22125108, 22121001, 92056104 and 22301250), the Natural Science Foundation of Fujian Province of China (2023J05005), the Fundamental Research Funds for the Central Universities (20720210002, 20720230003), and NFFTBS (J1310024).

## Author contributions

C.-T.L., L.-J.Q., and C.G. performed experiments. L.-G.L and X.L. performed DFT calculations. X.L., L.-W.Y., and B.Z. revised the paper. B.Z. conceived and directed the project and wrote the paper. All authors discussed the results and commented on the manuscript.

## Competing interests

The authors declare no competing interests.

## Additional information

**Peer review information** : *Nature Communications* thanks Baoguo Zhao and the other, anonymous, reviewer(s) for their contribution to the peer review of this work. A peer review file is available.

