## [Peer Review File · Nature Communications]

Asymmetric formal C–C bond insertion into aldehydes via copper-catalyzed diyne cyclizationReviewers' Comments:

Reviewer #1:

Remarks to the Author:

In this submitted article, Lu, Zhou and co-workers describe a strategy on a formal C-C bond insertion into aldehyde via Cu-catalyzed diyne cyclization. Subsequently, the reaction has been applied successfully in an asymmetric version, yielding axially chiral phosphine naphthylpyrroles with good to excellent enantioselectivity. Later, authors proved the reaction mechanism, origin of regio- and stereoselectivity by experimental and computational studies. Interestingly, axially chiral phosphine ligand was synthesized and successfully applied to asymmetric catalysis. Although a similar type of reaction yielding naphthylpyrroles is published recently (Angew. Chem. Int. Ed. 2023, 62, e202303670); however, the present reaction is conceptually different from the previous one and the final product is more complex aldehyde. Overall, the manuscript is well presented and has merit for publication in Nature Communication after minor revision.

Comments:

- 1) Table-2, E.e correct should be corrected to ee
- 2) Does Ms-protecting group not suitable in asymmetric version?
- 3) In reaction scope study (Fig 2), does this reaction works when R1 and R2 are alkyl groups or R2 is alkyl group?
- 4) please cite the relevant reference (Angew. Chem. Int. Ed. 2023, 62, e202303670)
- 5) ¹H NMR spectra of (3u, 3w) contains some impurities, please purify them

Reviewer #2:

Remarks to the Author:

This manuscript describes an asymmetric formal C-C bond insertion into aldehydes through diyne cyclization strategy with Cu(I)/SaBOX as the catalyst, providing a series of homologated pyrrolaldehydes and axially chiral naphthylpyrroles in moderate to good yields with good to high enantioselectivities. The key of the ingenious design is the vinyl cation intermediate generated from diyne cyclization which is used as the precursor of reactive carbonyl ylide with electron-rich substituents in replacement of diazo compounds. Moreover, the experimental and computational mechanistic studies were also conducted to reveal the reaction mechanism. This work offers an alternative protocol to achieve asymmetric formal C-C bond insertion into aldehydes utilizing non-diazo approach. The manuscript is well written and the experiments are carefully performed. Thus, I am convinced that this manuscript is of interest for the broad readership of Nat. Commun. The publication of this work in Nat. Commun. is recommended after authors addressing the following concerns and questions.

1. The diastereoselectivities of 5s-5u are still unsatisfied due to the challenge of construction of quaternary carbon stereocenter. Did the authors ever try to examine the formaldehyde to avoid the issue?
2. The naphthylpyrrole product 12 was transformed to an axially chiral phosphine ligand 14, which was used in [3+2] cycloaddition and allylic alkylation reaction respectively. It's suggested to provide the results of classic phosphine ligands for comparison.
3. The NMR spectra of 3s, 3u, 3w, 9 and 11 show signals that hint to some impurities. These compounds should be re-purified and the yields adjusted.

Reviewer #3:

Remarks to the Author:

The manuscript "Asymmetric formal C-C bond insertion into aldehydes via copper-catalyzed diyne cyclization" by Zhou and coworkers reports an alternative method for homologation of aldehyde that

does not require the more hazardous diazo compounds. In this work, the authors proposed a pathway that goes through a reactive carbonyl ylide intermediate, which in turn allows rapid rearrangement to the desired formal insertion product.

While I have not spotted any serious flaw in the experimental procedure, claiming the current approach in the abstract and introduction as a homologation of aldehyde sounds like an overstatement to me. The current approach seems to heavily rely on diyne cyclization to generate a crucial reactive intermediate and I am not convinced about the generalizability of this method. In my point of view, the value of the current work lies mostly on the realization of highly enantioselective axial chirality control.

Major issues:

While I consider the experimental part to be generally acceptable, I have serious concerns on the computational effort.

- The major claim on the origin of enantioselectivity is based on the computed TS with SaBOX ligand (L4), namely TS_A-(R) and TS_A-(S) in Fig. 7. There are, however, serious inconsistencies in the chirality of ligands presented in the manuscript. While the chirality of ligand L4 in the Lewis structure on the bottom-right of Fig. 7 is consistent with the chirality of L4 in Table 2, the chirality of L4 in the ball-and-stick structure on the top-right of Fig. 7 is EXACTLY OPPOSITE. This means the presented computational result actually supports the enantioselectivity OPPOSITE to what the authors have claimed, if the computational results are taken as-is.

- A more concerning problem was revealed when I checked the optimized coordinates in the hope of confirming the chirality issue. Specifically, the bridgehead carbon next to each oxygen atom of L4 in the computational models (both TS_A-(R) and TS_A-(S)) had only THREE as opposed to usual four bonds (i.e. one proton was missing). While authors have not provided the total charge of the computed system nor the computed electronic energies for me re-validate and determine the actual influence to the computed geometry, this error is anyway unacceptable and should render the whole computational results on enantioselectivity untrustable.

- Even if we disregard the stated issues for now, the whole catalytic cycle is not in line with the SaBOX transition states used to explain the enantioselectivities. While I do understand the possible limitation in computing power that calls for a simplified model, the computational model used in the catalytic cycle uses a monodentate ligand (MeCN) while the SaBOX ligand of interest serves as a bidentate ligand. As coordination number and coordination geometry plays a very important role in organometallic reaction mechanisms, simply replacing the bidentate SaBOX with monodentate MeCN is unlikely to give comparable results.

Taking all these into account, I do believe that the current computational section is doing more harm than good, to the point that I would suggest a rejection of the current manuscript unless the authors remove the computational parts from the whole work.

While I do value the successful construction of axially chiral skeleton, I think more support for the carbonyl ylide intermediate should be provided in order to pose that as a generalizable strategy. The current approach is highly limited to the specific diyne substrate that I am strongly hesitant to simply classify it as "carbene equivalent" (e.g. quoted in bottom left corner of Fig. 1) and claim to be a "formal C-C bond insertion strategy".

Minor issues:

- The authors should clearly indicate what [Cu] means in the manuscript, especially the charge it implies. If [Cu] is not a neutral species (which I supposed it is not), the [Cu]- notation in places like

the intermediate C in Fig 7 would be confusing.

- As both NMR and HPLC characterization are carried out on the mixture of isomers of 5u, which I assumed the authors have failed to separate the diastereomers, it is unclear to me how the correspondence of the 4 HPLC peaks are determined in order to calculate the presented dr and ee.

Typographical issue:

- page 11, line 159: "sliver-catalyzed" should be "silver-catalyzed"

廈門大學

XIAMEN UNIVERSITY
Xiamen, Fujian, China

I. Response to reviewer 1:

- (a) **Original comments:** *In this submitted article, Lu, Zhou and co-workers describe a strategy on a formal C-C bond insertion into aldehyde via Cu-catalyzed diyne cyclization. Subsequently, the reaction has been applied successfully in an asymmetric version, yielding axially chiral phosphine naphthylpyrroles with good to excellent enantioselectivity. Later, authors proved the reaction mechanism, origin of regio- and stereoselectivity by experimental and computational studies. Interestingly, axially chiral phosphine ligand was synthesized and successfully applied to asymmetric catalysis. Although a similar type of reaction yielding naphthylpyrroles is published recently (Angew. Chem. Int. Ed. **2023**, 62, e202303670); however, the present reaction is conceptually different from the previous one and the final product is more complex aldehyde. Overall, the manuscript is well presented and has merit for publication in Nature Communication after minor revision.*

We appreciate reviewer 1's kind support and excellent comments!

- (b) **Original comments:** *Table-2, E.e correct should be corrected to ee.*

This is a good suggestion. The statement mentioned has been corrected accordingly in the revised manuscript (Table 2).

- (c) **Original comments:** *Does Ms-protecting group not suitable in asymmetric version?*

We thank reviewer 1's great suggestion! The suggested substrate with Ms-protecting group (**4e**) was synthesized and tested in the asymmetric transformation. It was found that substrate **4e** demonstrated good reactivity under the standard reaction conditions, delivering axially chiral naphthylpyrrole **5e** in 89% yield with 86% ee. This result has been added into the revised manuscript (Fig. 3).

- (d) **Original comments:** *In reaction scope study (Fig 2), does this reaction works when R^1 and R^2 are alkyl groups or R^2 is alkyl group?*

We appreciate this constructive suggestion! The suggested experiments have been carried out and their results were included in the revised supporting information (Supplementary Table 3). Consistent with our previous reports on diyne cyclization (*J. Am. Chem. Soc.* **2019**, *141*, 16961; *J. Am. Chem. Soc.* **2020**, *142*, 7618), alkyl substituted ($R^2 = \text{alkyl}$) diynes **1** were found to be not compatible in the transformation and most of them gave inseparable mixtures. It's clear that substrates bearing electron-rich aryl groups gave better results. The possible reason is that these aryl groups ($R^2 = \text{aryl}$) play a key role in stabilizing the vinyl cation intermediates for further trapping.

廈門大學

XIAMEN UNIVERSITY
Xiamen, Fujian, China

- (e) **Original comments:** please cite the relevant reference (*Angew. Chem. Int. Ed.* 2023, 62, e202303670).

This is a great suggestion! Actually, the mentioned reference has been cited in our original manuscript (ref 59).

- (f) **Original comments:** ¹H NMR spectra of (3u, 3w) contains some impurities, please purify them.

We thank reviewer 1's great suggestions. The ¹H NMR spectra mentioned (3u, 3w) have been purified and their yields have also been adjusted. The corresponding spectra and yields were updated in the revised manuscript and supporting information.

II. Response to reviewer 2:

- (a) **Original comments:** This manuscript describes an asymmetric formal C - C bond insertion into aldehydes through diyne cyclization strategy with Cu(I)/SaBOX as the catalyst, providing a series of homologated pyrrolaldehydes and axially chiral naphthylpyrroles in moderate to good yields with good to high enantioselectivities. The key of the ingenious design is the vinyl cation intermediate generated from diyne cyclization which is used as the precursor of reactive carbonyl ylide with electron-rich substituents in replacement of diazo compounds. Moreover, the experimental and computational mechanistic studies were also conducted to reveal the reaction mechanism. This work offers an alternative protocol to achieve asymmetric formal C - C bond insertion into aldehydes utilizing non-diazo approach. The manuscript is well written and the experiments are carefully performed. Thus, I am convinced that this manuscript is of interest for the broad readership of *Nat. Commun.* The publication of this work in *Nat. Commun.* is recommended after authors addressing the following concerns and questions.

We appreciate reviewer 2's generous support!

廈門大學

XIAMEN UNIVERSITY

Xiamen, Fujian, China

(b) **Original comments:** *The diastereoselectivities of 5s-5u are still unsatisfied due to the challenge of construction of quaternary carbon stereocenter. Did the authors ever try to examine the formaldehyde to avoid the issue?*

We appreciate this suggestion! The suggested reaction using formaldehyde has been carried out, however no product was observed. The low nucleophilicity of formaldehyde could be the reason for the poor result. Moreover, systematic evaluation of more aryl aldehydes were conducted and most of these aryl aldehydes gave relatively low yields with 1:1-2:1 dr, which might due to the direct intermolecular attack of these nucleophiles onto ynamides. The suitable nucleophilicity of aldehyde is necessary for the atroposelective transformation. Thus, the diastereocontrol involving the quaternary carbon stereocenter is still challenging at this stage. These results have been included in the revised supporting information (Supplementary Table 4).

(c) **Original comments:** *The naphthylpyrrole product 12 was transformed to an axially chiral phosphine ligand 14, which was used in [3+2] cycloaddition and allylic alkylation reaction respectively. It's suggested to provide the results of classic phosphine ligands for comparison.*

We thank reviewer 2's wonderful suggestions! Most of the reported enantioselective [3+2] cycloaddition and allylic alkylation rely on bisphosphine ligands (For selected examples on silver-catalyzed [3+2] cycloaddition, see: *J. Am. Chem. Soc.* **2012**, *134*, 12936; *Org. Lett.*

廈門大學

XIAMEN UNIVERSITY

Xiamen, Fujian, China

2003, 5, 5043. For selected examples on palladium-catalyzed allylic alkylation, see: *Angew. Chem. Int. Ed.* **2019**, 58, 4714; *Chem. Commun.* **1999**, 1895).

As suggested, classic phosphine ligands were selected based on above references, and the target reactions have been conducted for comparison. As for [3+2] cycloaddition, our axially chiral monophosphine ligand **14** demonstrated better reactivity and higher enantioselectivity than bisphosphine ligand (*S,S*)-Ph-BPE and monophosphine ligand (*S*)-*N*-^{*i*}Pr₂-MonoPhos. As for allylic alkylation, our axially chiral monophosphine ligand **14** showed higher reactivity and lower enantioselectivity than bisphosphine ligand (*S,S*)-Ph-BPE and monophosphine ligand (*S*)-*N*-^{*i*}Pr₂-MonoPhos. Therefore, our axially chiral monophosphine ligand **14** or its skeleton is potentially useful in asymmetric catalysis. These results have been included into the revised manuscript (Fig. 5).

- (d) **Original comments:** *The NMR spectra of 3s, 3u, 3w, 9 and 11 show signals that hint to some impurities. These compounds should be re-purified and the yields adjusted.*

We thank reviewer 2's suggestion. Compounds **3s**, **3u**, **3w**, **9** and **11** have been purified accordingly. Their updated NMR spectra and yields were included into the revised manuscript and supporting information.

III. Response to reviewer 3:

- (a) **Original comments:** *The manuscript "Asymmetric formal C - C bond insertion into aldehydes via copper-catalyzed diyne cyclization" by Zhou and coworkers reports an alternative method for homologation of aldehyde that does not require the more hazardous diazo compounds. In this work, the authors proposed a pathway that goes through a reactive carbonyl ylide intermediate, which in turn allows rapid rearrangement to the desired formal insertion product.*

While I have not spotted any serious flaw in the experimental procedure, claiming the current approach in the abstract and introduction as a homologation of aldehyde sounds like an overstatement to me. The current approach seems to heavily rely on diyne cyclization to generate a crucial reactive intermediate and I am not convinced about the generalizability of this method. In my point of view, the value of the current work lies mostly on the realization of highly enantioselective axial chirality control.

We appreciate reviewer 3's kind comments and generous support on our work! Indeed, this atroposelective transformation relies on the unique vinyl cation intermediate generated from diyne cyclization, enabling the preparation of axially chiral naphthylpyrroles. Further investigation into simpler starting materials and intermolecular reactions are ongoing in our laboratory.

- (b) **Original comments:** *While I consider the experimental part to be generally acceptable, I have serious concerns on the computational effort.*

- The major claim on the origin of enantioselectivity is based on the computed TS with SaBOX ligand (L4), namely TS_A-(R) and TS_A-(S) in Fig. 7. There are, however, serious

廈門大學

XIAMEN UNIVERSITY
Xiamen, Fujian, China

inconsistencies in the chirality of ligands presented in the manuscript. While the chirality of ligand L4 in the Lewis structure on the bottom-right of Fig. 7 is consistent with the chirality of L4 in Table 2, the chirality of L4 in the ball-and-stick structure on the top-right of Fig. 7 is EXACTLY OPPOSITE. This means the presented computational result actually supports the enantioselectivity OPPOSITE to what the authors have claimed, if the computational results are taken as-is.

We strongly appreciate reviewer 3's careful and thorough inspections, which helped us a lot to improve the computational studies. We apologize for the serious mistakes on chiral ligand **L4** and the attached XYZ coordinates in our computational results during editing. To correct these errors, we recalculated the whole reaction and double-checked the chirality of ligand and the enantioselectivity of reaction (the calculations supported our observed enantioselectivity well). These results have been included into the revised manuscript (Fig. 7) and supporting information (Supplementary Figure 1-2).

- (c) **Original comments:** - *A more concerning problem was revealed when I checked the optimized coordinates in the hope of confirming the chirality issue. Specifically, the bridgehead carbon next to each oxygen atom of L4 in the computational models (both TS_A-(R) and TS_A-(S)) had only THREE as opposed to usual four bonds (i.e. one proton was missing). While authors have not provided the total charge of the computed system nor the computed electronic energies for me re-validate and determine the actual influence to the computed geometry, this error is anyway unacceptable and should render the whole computational results on enantioselectivity untrustable.*

We apologize for the serious mistakes for chiral ligand **L4** and the attached XYZ coordinates in our computational studies during editing. To correct these mistakes, we recalculated the whole reaction and double-checked the structure of ligand and the enantioselectivity of reaction.

Additionally, the total charge of the computed system and computed electronic energies have all been added in the updated computational results. These results have been included into the revised manuscript (Fig. 7) and supporting information (Supplementary Figure 1-2).

- (d) **Original comments:** - *Even if we disregard the stated issues for now, the whole catalytic cycle is not in line with the SaBOX transition states used to explain the enantioselectivities. While I do understand the possible limitation in computing power that calls for a simplified model, the computational model used in the catalytic cycle uses a monodentate ligand (MeCN) while the SaBOX ligand of interest serves as a bidentate ligand. As coordination number and coordination geometry plays a very important role in organometallic reaction mechanisms, simply replacing the bidentate SaBOX with monodentate MeCN is unlikely to give comparable results.*

Taking all these into account, I do believe that the current computational section is doing more harm than good, to the point that I would suggest a rejection of the current manuscript unless the authors remove the computational parts from the whole work.

廈門大學

XIAMEN UNIVERSITY
Xiamen, Fujian, China

We thank this excellent question! We agree that the simplified model using MeCN as ligand could give different results compared with the bidentate SaBOX. To be responsible for our study, we recalculated the whole system using SaBOX as ligand and double-checked all the results (the calculations supported our enantioselective reaction well, Fig. 7).

To be consistent with the whole work, we wish to include the new computational studies in the revised manuscript and supporting information (as highlighted in the revised manuscript and SI), if the reviewer agrees. Otherwise, we can put them only into supporting information.

- (e) **Original comments:** *While I do value the successful construction of axially chiral skeleton, I think more support for the carbonyl ylide intermediate should be provided in order to pose that as a generalizable strategy. The current approach is highly limited to the specific diyne substrate that I am strongly hesitant to simply classify it as “carbene equivalent” (e.g. quoted in bottom left corner of Fig. 1) and claim to be a “formal C-C bond insertion strategy”.*

We thank reviewer 3's intriguing suggestions! We agree with reviewer 3 for the carbonyl ylide intermediate and believe it's the key for the transformation. While carbonyl ylide intermediate is hard to be directly obtained through experimental approaches, our updated computational studies have been conducted, which supported the formation of carbonyl ylide intermediate (or its resonance form) with low energy barrier.

To avoid misunderstanding, we changed the statement of vinyl cation from “carbene equivalent” to “carbene-like intermediate” in Fig. 1 (For related statement of “carbene-like” on vinyl cations, see: *Angew. Chem. Int. Ed.* **2018**, 57, 16942; *Sci. China Chem.* **2022**, 65, 20). Indeed, the current approach is still limited to diyne substrates and further investigations into simpler precursors and intermolecular reactions are ongoing. From the view of aldehydes (both substrates and products), this reaction could be considered as the formal C-C bond insertion into aldehydes based on carbene-like intermediate (vinyl cation), although with limitations. The statement mentioned has been changed accordingly in the revised manuscript (Fig. 1 and related text).

- (f) **Original comments:** *- The authors should clearly indicate what [Cu] means in the manuscript, especially the charge it implies. If [Cu] is not a neutral species (which I supposed it is not), the [Cu]- notation in places like the intermediate C in Fig 7 would be confusing.*

We apologize for the misunderstanding caused by our statement. In our original manuscript, [Cu] represents a ligated cooper species with a positive charge. To avoid confusions, we have corrected [Cu] to Cu⁺L4 and [Cu] to CuL4 in the revised manuscript (Fig. 7) and supporting information (Supplementary Figure 1).

- (g) **Original comments:** *As both NMR and HPLC characterization are carried out on the mixture of isomers of 5u, which I assumed the authors have failed to separate the*

廈門大學

XIAMEN UNIVERSITY
Xiamen, Fujian, China

diastereomers, it is unclear to me how the correspondence of the 4 HPLC peaks are determined in order to calculate the presented dr and ee.

We thank reviewer 3's great comments. Extensive efforts have been made to separate the diastereomers of compound **5v**, but they cannot be isolated well (the number of **5u** was changed to **5v** in the revised manuscript, because of the incorporation of a new Ms-protected example **5e** according to another reviewer's comments). After multiple attempts, we got the diastereomers of **5v** with higher dr (2.5:1), which gave clearer NMR and HPLC spectra. These NMR and HPLC spectra have been included into the revised supporting information.

Here are the details for the calculation of dr and ee:

(1) The dr of **5v** was calculated from the analysis into crude ^1H NMR of reaction mixture before purification. As shown below, the integration in crude ^1H NMR of reaction mixture shows a ratio of 1.4:1 for diastereomers, and this dr ratio is consistent with the ^1H NMR for the mixture of diastereomers after quick column chromatography.

廈門大學

XIAMEN UNIVERSITY
Xiamen, Fujian, China

(2) The ee values were calculated from the 4 HPLC peaks and these peaks could be well-assigned to two diastereomers of **5v**. As shown below, integration of HPLC for the mixture of diastereomers shows that the first and third peaks belong to major diastereomer (relative area 1:1) and the second and fourth peaks belong to minor diastereomer (relative area 1:1). According to the assignment of peaks and retention time for racemic compounds, the ee values of chiral diastereomers were calculated to be 98% (major) and 97% (minor). Also, the spectra of diastereomers with higher dr (2.5:1, as mentioned above) further confirmed the assignment of the 4 HPLC peaks and their ees (shown in revised supporting information).

廈門大學

XIAMEN UNIVERSITY
Xiamen, Fujian, China

(h) Original comments: page 11, line 159: “sliver-catalyzed” should be “silver-catalyzed”

We apologize for the typo in the manuscript. The statement suggested has been changed accordingly in the revised manuscript.

Reviewers' Comments:

Reviewer #1:

Remarks to the Author:

All the concerns raised by me during revision have been addressed successfully by the author and now the revised version could be accepted for publication in Nature Communication.

Reviewer #2:

Remarks to the Author:

The revision of the manuscript pertinently addresses the reviewers' previous comments. Examining the substrate scope of other diyenes and aldehydes, NMR of re-purified products, and the results of recalculation of the reaction were all supplemented in the revised manuscript and SI. The revisions made the description more accurate. Given those improvements, I would suggest the publication of this manuscript in Nat. Commun.

Reviewer #3:

Remarks to the Author:

In this revised manuscript, the authors did show their great effort in improving the manuscript, by performing additional experiments, and more notably, redo all the computations correcting all the fatal mistakes (e.g. wrong chirality and missing proton) in the previous version. I do appreciate the amount of input by the authors within the tight time frame.

While most technical errors of the computational results have been resolved, the presentation of the catalytic cycles is still not fully satisfactory.

(1) While the catalytic cycle appeared to be "closed" at the top left hand corner of Fig. 7, the presentation of states are actually problematic. For the current representation, it appears to me that Cu(L)4 is used in stoichiometric amount according to the current presentation of catalytic cycle, but in fact it is used catalytically.

If my understanding is correct, the Cu(L)4 to Cu(L4) conversion is only performed once, but the 4f to 5f conversion is carried out repeatedly. Drawing an arrow that connects "Cu(L)4 + (L4) + 4f" to "Cu(L4) + 4L + 5f" is clearly mixing the two different situations.

The TS_F transition state that seems to lie in the middle is even more problematic. While due to the aforementioned reason, it clearly does not connect "Cu(L)4 + (L4) + 4f" to "Cu(L4) + 4L + 5f". At the same time, it doesn't even connect F and A, because that would involve the exchange between new substrate 4f and the resulting product 5f. This exchange transition state is not modelled in the DFT calculation either.

(2) Again for the process between F and A, stating the energies on the state would lead to the illusion that F->A is energetically unfavourable (because the relative Gibbs free energy goes from -56.4 to -35.1), but I supposed it is not the case (otherwise the whole catalytic reaction would not occur spontaneously). It would be more appropriate to draw the free energy landscape of one catalytic cycle (from A to A), following the standard practice in the field, so that readers can appreciate not just the free energy change in individual steps, but also the free energy gain over one cycle (which is not shown in the current presentation).

(3) The process 1,4-H migration and demetallation is simply not shown in the calculation and is only

briefly stated in the catalytic cycle. However, considering the great change in the Cu position, it is hard to believe the process is trivial enough to be safely omitted.

(4) Authors should double check the accuracy of presentation of the mechanistic cycle. There are still inconsistencies between the presented state and the computational results. For example, it is shown in the catalytic cycle that $B + 2h \rightarrow C$, but B and C actually contains the same number of atoms in the computational results (because state "B" already contains 2h).

(5) I personally still have doubts on the state F and TS_F, considering the atypical bond transformations. Though I currently do not have alternative suggestions, and would reconsider the situation after the authors have addressed the issue in (2).

廈門大學
XIAMEN UNIVERSITY
Xiamen, Fujian, China

I. Response to reviewer 1:

- (a) **Original comments:** *All the concerns raised by me during revision have been addressed successfully by the author and now the revised version could be accepted for publication in Nature Communication..*

We appreciate reviewer 1's kind support and excellent comments!

II. Response to reviewer 2:

- (a) **Original comments:** *The revision of the manuscript pertinently addresses the reviewers' previous comments. Examining the substrate scope of other diyenes and aldehydes, NMR of re-purified products, and the results of recalculation of the reaction were all supplemented in the revised manuscript and SI. The revisions made the description more accurate. Given those improvements, I would suggest the publication of this manuscript in Nat. Commun..*

We thank reviewer 2's generous support for improving our work!

III. Response to reviewer 3:

- (a) **Original comments:** *In this revised manuscript, the authors did show their great effort in improving the manuscript, by performing additional experiments, and more notably, redo all the computations correcting all the fatal mistakes (e.g. wrong chirality and missing proton) in the previous version. I do appreciate the amount of input by the authors within the tight time frame.*

We appreciate reviewer 3's generous support! The suggestions from reviewer 3 helped us a lot to improve our work.

- (b) **Original comments:** *While most technical errors of the computational results have been resolved, the presentation of the catalytic cycles is still not fully satisfactory.*

(1) While the catalytic cycle appeared to be "closed" at the top left hand corner of Fig. 7, the presentation of states are actually problematic. For the current representation, it appears to me that Cu(L)4 is used in stoichiometric amount according to the current presentation of catalytic cycle, but in fact it is used catalytically.

If my understanding is correct, the Cu(L)4 to Cu(L4) conversion is only performed once, but the 4f to 5f conversion is carried out repeatedly. Drawing an arrow that connects "Cu(L)4 + (L4) + 4f" to "Cu(L4) + 4L + 5f" is clearly mixing the two different situations.

The TS_F transition state that seems to lie in the middle is even more problematic. While due to the aforementioned reason, it clearly does not connect "Cu(L)4 + (L4) + 4f" to "Cu(L4) + 4L + 5f". At the same time, it doesn't even connect F and A, because that would involve the exchange between new substrate 4f and the resulting product 5f. This exchange transition state is not modelled in the DFT calculation either.

We appreciate reviewer 3's great suggestions. Indeed, the copper complex was used catalytically. To avoid misunderstanding, we changed the picture of catalytic cycle. These

廈門大學

XIAMEN UNIVERSITY

Xiamen, Fujian, China

results have been included into the revised manuscript (Fig. 7) and SI (Supplementary Figure 1-2).

- (c) **Original comments:** (2) *Again for the process between F and A, stating the energies on the state would lead to the illusion that F->A is energetically unfavourable (because the relative Gibbs free energy goes from -56.4 to -35.1), but I supposed it is not the case (otherwise the whole catalytic reaction would not occur spontaneously). It would be more appropriate to draw the free energy landscape of one catalytic cycle (from A to A), following the standard practice in the field, so that readers can appreciate not just the free energy change in individual steps, but also the free energy gain over one cycle (which is not shown in the current presentation).*

We thank reviewer 3's helpful suggestion. The mentioned catalytic cycle has been changed and included into the revised manuscript (Fig. 7) and SI (Supplementary Figure 1-2).

- (d) **Original comments:** (3) *The process 1,4-H migration and demetallation is simply not shown in the calculation and is only briefly stated in the catalytic cycle. However, considering the great change in the Cu position, it is hard to believe the process is trivial enough to be safely omitted.*

This is a good question! As suggested, we recalculated 1,4-H migration and demetallation process. During the calculation, we found the direct 1,4-H migration without any mediator has a very high energy barrier. Inspired by our previous work involving Lewis base-assisted 1,4-H migration (*Angew. Chem. Int. Ed.* **2023**, 62, e202303670; *Chem. Sci.* **2021**, 12, 9466), we finally found aldehyde could act as Lewis base to promote the 1,4-proton transfer step (energy barrier 18.7 kcal/mol) and following demetallation step has a very low energy barrier according to our previous study.

To be more accurate, we independently calculated the catalytic cycle of desired product with two opposite configurations ((*R*)-**5f**, major and (*S*)-**5f**, minor). Consistent with previous results, the diyne cyclization is the enantio-determining step (from intermediate **A** to **B**), and ring-opening of 1,3-dioxolane is the rate-determining step (from intermediate **F** to **G**). These results have been included into the revised manuscript (Fig. 7) and SI (Supplementary Figure 1-2).

廈門大學

XIAMEN UNIVERSITY
Xiamen, Fujian, China

- (e) **Original comments:** (4) Authors should double check the accuracy of presentation of the mechanistic cycle. There are still inconsistencies between the presented state and the computational results. For example, it is shown in the catalytic cycle that $B + 2h \rightarrow C$, but B and C actually contains the same number of atoms in the computational results (because state "B" already contains 2h).

We thank reviewer 3's thorough inspections! We've recalculated the intermediate **B** according to the suggestions, and double-checked our updated mechanistic cycle. These results have been included into the revised manuscript (Fig. 7) and SI (Supplementary Figure 1-2).

- (f) **Original comments:** (5) I personally still have doubts on the state F and TS_F , considering the atypical bond transformations. Though I currently do not have alternative suggestions, and would reconsider the situation after the authors have addressed the issue in (2).

We thank reviewer 3's suggestion. Further DFT calculations have been carried out to figure out other possible cationic intermediates and transition states (such as the related oxonium intermediate shown below), however we cannot locate those intermediates and transition states. The pathway through transition state F (the number changed to G in updated manuscript) shows the best result currently.

Reviewers' Comments:

Reviewer #3:

Remarks to the Author:

I appreciate the authors' effort and have no further comments or questions.

廈門大學
XIAMEN UNIVERSITY
Xiamen, Fujian, China

I. Response to reviewer 3:

(a) **Original comments:** *I appreciate the authors' effort and have no further comments or questions.*

We strongly appreciate reviewer 3's generous support and helpful discussion! These suggestions helped us a lot to improve our work.